# Good Statistical Practices in Agronomy Using Categorical Data Analysis, with Alfalfa Examples Having Poisson and Binomial Underlying Distributions

Ronald P. Mowers [1,*], Bruna Bucciarelli [2], Yuanyuan Cao [3,4], Deborah A. Samac [3,5] and Zhanyou Xu [2,3,*]

[1] Vis Viva Energy Economics Consulting, 2114 State Ave., Ames, IA 50014, USA
[2] Department of Agronomy and Plant Genetics, University of Minnesota, 1991 Upper Buford Circle, St. Paul, MN 55108, USA; bucci002@umn.edu
[3] USDA-ARS, Plant Science Research Unit, 1991 Upper Buford Circle, St. Paul, MN 55108, USA; yy721@hotmail.com (Y.C.); debby.samac@usda.gov (D.A.S.)
[4] School of Life Sciences, Anhui Agricultural University, Hefei 230036, China
[5] Department of Plant Pathology, University of Minnesota, 1991 Upper Buford Circle, St. Paul, MN 55108, USA
[*] Correspondence: ronpmowers@gmail.com (R.P.M.); zhanyou.xu@usda.gov (Z.X.)

**Abstract:** Categorical data derived from qualitative classifications or countable quantitative data are common in biological scientific work and crop breeding. Categorical data analyses are important for drawing correct inferences from experiments. However, categorical data can introduce unique issues in data analysis. This paper discusses common problems arising from categorical variable analysis and modeling, demonstrates the issues or risks of misapplying analysis, and suggests approaches to address data analysis challenges using two data sets from alfalfa breeding programs. For each data set, we present several analysis methods, e.g., simple t-test, analysis of variance (ANOVA), split plot analysis, generalized linear model (glm), generalized linear mixed model (glmm) using R with R markdown, and with the standard statistical analysis software SAS/JMP. The goal is to demonstrate good analysis practices for categorical data by comparing the potential 'bad' analyses with better ones, avoiding too much reliance on reaching a significant *p*-value of 0.05, and navigating the morass of ever-increasing numbers of potential R functions. The three main aspects of this research focus on choosing the right data distribution to use, using the correct error terms for hypothesis test *p*-values including the right type of sum of the squares (Type I, II, and III), and proper statistical models for categorical data analysis. Our results show the importance of good statistical analysis practice to help agronomists, breeders, and other researchers apply appropriate statistical approaches to draw more accurate conclusions from their data.

**Keywords:** categorical data analysis (CDA); Poisson; binomial; generalized linear model (glm); generalized linear mixed model (glmm)

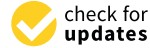



## 1. Introduction

The purpose of this article is to show statistical issues faced in the analysis of experiments with categorical data and approaches to solve these issues. We highlight some faulty analyses and demonstrate other, better solutions. Although the statistical methods presented here are not new, this guide for practitioners provides two relevant examples and presents categorical data analysis (CDA) methods not usually given in initial statistics courses for agronomists.

Explanations for each example begin with an outline of the approach used to analyze the data, show issues which arise in data analyses for the two alfalfa (*Medicago sativa* L.) example data sets, and help readers understand good statistical practices to complete solutions and draw conclusions. The two examples are a nodules per root experiment (N/R experiment) with underlying Poisson distribution and a phosphorous winter-survival

experiment (P_WS experiment) as an example of data having a binomial distribution. Throughout the analyses, we compare results from the programming language R with output from JMP, a Statistical Analysis Systems (SAS Institute, 2017, JMP software, Version 13, SAS Inst., Cary, NC, USA) [1,2] product which has an emphasis on interactive statistical graphics.

The characteristics of CDA are shown in the context of building upon a previous statistics course that many agronomists and plant breeders take in graduate school. Such a course usually assumes observations from normal, independent, identically distributed (NID) random variables and teaches analysis of variance (anova) and multiple linear regression. This article describes modifications when distributions differ from the NID assumptions because underlying distributions are Poisson or binomial. The approach used in analyzing data for each example is to first use the more familiar linear models with NID assumptions, next use data transformations or averages to improve these linear models, and finally to use generalized linear models (glms) to better account for response variables which do not follow the normal distribution. Other potential pitfalls are presented in the analyses of these experiments and ways to avoid pitfalls and perform proper analyses are shown.

The first question is: "Why use CDA?" The normal distribution is very important in statistical analyses because it has well-established theory for estimating treatment effects or testing hypotheses in designed experiments, it is the distribution that sample means from other distributions approach as sample size increases, and there are many good examples of traits, such as crop yield and plant height, which seem to fit this distribution well ([3], p. 40). However, there are other important distributions in agronomy and plant breeding which may not at all be well described by the normal distribution, such as proportion data from a binomial distribution, e.g., winter-survival yes-no binary data. Agresti notes that if sample sizes are not large, a normal probability model might predict proportions outside the range of zero to one, giving a nonsensical result, such as a survival rate of 103% ([4], p. 68). Agresti also states that effects of explanatory variables on proportion, p, are usually nonlinear, with the log-odds ratio, $\log[p/(1-p)]$, more likely to be linear in the explanatory variables. If sample size is large, the normal distribution can be used to approximate the binomial, Snedecor and Cochran giving the rule of thumb that if np and $n(1-p)$ are both greater than 15, the normal approximation is warranted ([3], p. 130). If the binomial proportion is really small, e.g., with very rare events, they state that the Poisson distribution becomes a more appropriate model. The Poisson distribution has mean and variance equal and is often used to model counts, such as the counts of nodules on the roots of an alfalfa plant ([4], pp. 72–76). Agresti states that such count data often have more variation than expected with a true Poisson distribution. The direct relation of mean and variance for Poisson in counts has a modeling advantage compared with the NID assumption when effects are proportional with proportion constant across treatments, e.g., one cultivar with 22% more nodules than another.

Categorical data analysis methods are discussed in books such as that by Agresti [4]. The best methods for analyzing these data are continually evolving. Before the advent of software for generalized linear models, standard practice was to transform the data to achieve distributions with variance more independent of the mean ([3], pp. 286–291). Often a first analysis can be performed with a normal approximation. One transformation for data whose variance is proportional to mean, e.g., Poisson distributed, such as counts of rare events, is to take the square root of the original data, then run analyses the in same way as for the NID assumptions. Proportion data traditionally were transformed with the arcsine(sqrt(p)). If the standard deviation in the original scale varies directly as the mean, i.e., the coefficient of variation (CV) is constant, data can be log transformed before running the usual analyses. This not only helps equalize variance for values, but helps achieve additivity of effects that are proportional in the original scale. In 1972 Nelder and Wedderburn proposed a better method for analyzing non-normal data using a generalized linear model (glm; plural glms) [5]. These glms give more precise analyses for the count

and other non-normal data by utilizing the actual distributions, such as the binomial or Poisson [6]. This method can be used even for normally distributed variables by applying an identity link function [4]. The method uses a linear predictor of explanatory variables, such as replicate and treatment, with a link function of the mean. For binomial data, the usual link function is the log odds ratio or logit, defined earlier, and for Poisson, it is a log link. Software to fit these models has only been readily available since the late 1990s. Stroup [6] recommended use of glms rather than the historical logarithmic or square root transformations formerly used to model count data. JMP and SAS software (now PROC GLIMMIX) have routines for analyzing glms, and in R, the glm and glmmTMB functions are available [2,7,8].

Our objectives are:

1. To raise awareness of categorical data analysis methods.
2. To demonstrate proper CDA analyses.
3. To highlight potential pitfalls and corrections in statistical analyses for the example data sets.

## 2. Literature Review

Some examples of pitfalls in running analyses of agronomic data which we address are:

**Pitfall 1. Using too simple an analysis, e.g., a simple t-test when we need to include other recognizable sources of variation in the model.** There is always a delicate balance between over-simplifying and including too many independent variables in a statistical model. James et al. discuss the bias-variance trade-off in statistical modeling, stating that neither over-simplified nor too-complicated models (which can even over-fit to the point of zero error degrees of freedom) are good for prediction ([9], pp. 33–36). For example, suppose we have an experiment with two cultivars and three other treatments in a $2 \times 3$ factorial. An over-simplified initial t-test can give a misleading first impression. All sources of variation other than that due to cultivar would be grouped into the error variance estimate, but important factors such as the other treatment and perhaps a blocking factor should have been recognized and not included in residual error. Without these sources of variation accounted for in the model, the error term would be greatly inflated, a major pitfall. The relation $F = t^2$ should hold for analysis of variance with a single two-level factor ([3], p. 224), and in this instance, such a single-factor anova model would be faulty because important other factors would be left out. Failure to include all important terms makes an overly-simple model less flexible, potentially resulting in large prediction bias [9].

**Pitfall 2. Using the wrong error term for desired inference.** A common mistake in analyses with "cookbook" methods is to use an error term having inflated degrees of freedom to test for statistical significance. For example, in a cultivar yield trial experiment with eight locations, each having a randomized complete block experiment, we generally wish to infer results to a new environment or set of environments. Running a linear model with location, cultivar, and location-cultivar interaction is reasonable, but using the residual error term from this model to test for cultivar differences can be misleading because we want to infer results not just to the exact environments tested, but to new ones, potentially in a future year or on a different field of which the environments are a sample ([2], p. 12). One approach which avoids the problem of having the wrong error term is to first obtain cultivar averages or least squares means for each location and then use these means in an across-locations analysis with cultivar and location as terms in the model [10]. Mohring and Piepho call this a two-stage analysis. These two-stage analyses typically require using standard errors from the first stage for weighting the averages in the second stage, but equal variances are often assumed when all first-stage means have the same or nearly the same numbers of observations. Split-plot experiment designs require the use of models with complicated error terms ([11], pp. 174–175). Stroup discusses the error structure in split-plot experiments, including an error component for whole-plots and another for subplots ([6], p. 821). Crawley even discusses a three-level split-split plot example ([11], p. 174). Expected values of mean squares guide researchers on error terms for hypothesis

tests in these split-plot designs ([3], pp. 322–323). Automatic tests of treatment effects in nested designs may differ in R functions from those in SAS and JMP, and it is important to realize if Type I, Type II, or Type III sums of squares are being used for these tests. The Type I, or sequential sums of squares in R functions, are used in aov and anova, for example in the command anova(lm(y~A*B, data = D), and differ from the Type III sums of squares and mean squares in the function Anova, for distinction written with capital A, in the R car package [12,13]. Type III sums of squares are the default in JMP [1]. The Type I sums of squares are helpful to analyze split-plot experiments, as Crawley illustrates in the R function aov ([11], p. 174). For a split-plot design with whole-plot factor A and subplot factor B, both nested in rep R, the split-plot analysis command is:

$$aov(y{\sim}A*B + Error(R + R{:}A), data = D). \tag{1}$$

An equivalent statement is aov(y~A*B + Error(R/A), data = D).

**Pitfall 3. Placing too much emphasis on achieving probabilities less than 0.05 in hypothesis tests.** Scientists and other practitioners seem obsessed with obtaining statistically significant test results, generally meaning finding a probability below 0.05 for their particular hypothesis test. Not only is this a dangerous practice in which results which are important but show no statistically significant difference in, for example, treatment versus control are not even published, but also many times the real point of the experiment is missed. For example, we may want to know the optimum phosphorous fertilizer rate for maximizing alfalfa plant over-winter survival, but simple hypothesis tests of each treatment versus control might not even be statistically significant when there is possibly a true quadratic relationship.

**Pitfall 4. Confusion in choosing the correct software function to analyze data.** There are so many available functions in a programming language such as R that it is hard for scientists to decide which to use. The number of available R packages to download is over 10,000 [14]. This makes it difficult for practitioners to decide which programs or subroutines to use when writing a program to analyze their data. Examples of well-vetted R functions are available in good textbooks which describe statistical procedures and give examples which illustrate how to use commands; for example, Agresti [4] and James at all [9]. Another guiding principle is to always check with your statistician, who is an expert in the area of data analyses, in the same way you would check with a plant pathologist if you found a new unfamiliar disease on your crop of interest.

### 3. Examples to Illustrate Good Statistical Practices in Categorical Data

**Outline of analysis of our two alfalfa experiments, the nodules per root (N/R) and phosphorus-winter survival (P_WS).** The structured approach to the analysis of data from these experiments includes determining the objective, detailing the experiment design, carefully determining quantities measured, specifying analyses and potential analysis issues, and finally making conclusions and communicating results. We give this outline for each of the experiments, first with the alfalfa nodules/root (N/R) experiment and next with the phosphorous winter survival (P_WS) experiment.

**Example 1. Nodules/root (N/R) experiment to compare alfalfa cultivars and nodulation strains**

**Ex1.1. Objective.** Objectives of the alfalfa nodules study were: (1) to test whether there is a difference in mean number of nodules on roots between two alfalfa cultivars, one with branching or fibrous root and the other with taproot architecture; (2) to test whether there are differences for different nodulation strains.

**Ex1.2. Experiment Design.** The design of this experiment is a randomized complete block for six treatments in a (2 × 3) factorial, two cultivars each having three inoculation treatments (two nodulation strains plus a control). The three blocks are repeats of greenhouse runs undertaken at different dates (called Times), with eight plants per treatment, each grown in separate pots, all completely randomized within the greenhouse. This results

in $3 \times 2 \times 3 \times 8 = 144$ total plants. However, six plants are missing due to plant death, so the total number of plants on which measurements are made is 138.

**Ex1.3. Data.** Nodules are counted on the roots of each plant after plants have been established and grown for 14 days (original data in supplemental file "nodules.csv" for detail). Nodule numbers range from 0 to 78, with 4 of the plants having 0 nodules. We note that the counts are likely to have a higher variance for higher mean count numbers, a potential issue for the data analysis if we use the NID assumptions. This type of count data are often considered to be Poisson-distributed [3,15].

**Ex1.4. Statistical Analysis.** The first analysis for this experiment uses the nodule count data (Y) in a linear model assuming normal independent distribution (NID), an analysis familiar to many scientists. Models in these statistical analysis subsections are written in the same way for both JMP and R, except for interactions (in JMP the A-B interaction is depicted as A*B and in R as A:B. In R, A*B means A + B + A:B). The model, written in R, is:

$$Y = \text{Time} + \text{Cultivar} * \text{Inoculum} + \text{Time:Cultivar} + \text{Time:Inoculum} + \text{Time:Cultivar:Inoculum} \tag{2}$$

where Time (=greenhouse run considered as a Rep), its interactions are assumed random effects, and Cultivar and Inoculum are fixed.

The second analysis uses the square-root data transformation and averages over pots to obtain an anova to test the null hypothesis of equal cultivar means and to estimate cultivar and inoculum treatment means, as well as to test the cultivar-inoculum interaction. Its model, written in JMP, is:

$$\text{Avg(sqrt(Y))} = \text{Time} + \text{Cultivar} + \text{Inoculum} + \text{Cultivar} * \text{Inoculum}, \tag{3}$$

where all factors, including the nuisance factor, Time, are considered fixed.

The third analysis has a generalized linear mixed model for Poisson distributed Y, with a model equation similar to that in Equation (2), except using glmer (see R Markdown supplement), and for this glmm, there is a normal random effect for pot to account for this recognized source of variation. This model also has omitted terms not statistically significant.

$$Y = \text{Time} + \text{Cultivar} + \text{Inoculum} + \text{Pot}, \tag{4}$$

where Time and Pot are random and other factors are fixed. This equation refers to modeling the standard Poisson log link function on these explanatory variables.

There are several issues related to these statistical analyses. We examine proper error terms for testing treatments in this design for different scopes of inference. Count data do not fit the usual anova normal independently distributed (NID) assumptions, and we have categorical data analysis software in R and JMP available for glm analyses for the Poisson distributed count data. We used the glmer function to fit the glmm (4) with log link and independent normal pot error term. The R codes can be found in a R markdown file "nodules.rmd", and the line-by-line results can be found in the supplemental file "nodules.docx" for details.

**Ex1.5. Interpretation.** A difference in numbers of nodules per plant may be associated with higher nitrogen (N) fixation, and it is important to find if a difference in nodules exists. A 95% confidence interval (CI) for the cultivar mean nodule difference would be valuable scientific information. These results might influence the direction of an alfalfa breeding program.

**Example 1 Results of Nodules/Root (N/R) Experiment**

**Over-simplified analyses can give misleading first impressions.** A researcher had requested a simple independent-sample t-test for cultivar difference. An issue with this test is that the *p*-value for cultivar difference from this did not agree with probabilities from a linear model analysis-of-variance. The anova table of the nodule counts, using Equation (2), is shown in Table 1. This table has a cultivar null hypothesis probability of

0.026, which differed from the independent sample t-test with probability of 0.19. The independent-sample t-test for cultivar differences was shown to be equivalent to a one-factor (cultivar) anova (see Figure 1), as discussed in the literature review. This example illustrates that the independent sample t-test comparing cultivars, because it does not take account of important sources of variation, inflates the error term. These sources of variation include blocks (greenhouse runs) and inoculum treatments, which, when part of the residual, makes the error term too high for the independent sample t-test compared with the multi-factor anova model.

**Table 1.** ANOVA table for nodule numbers, assumed with a normal distribution, from R code executed with multi-way ANOVA analysis.

| Terms * | Df | Sum Square | Mean Square | F Value | *p* Value | Exp (MS) |
|---|---|---|---|---|---|---|
| Time (T) | 2 | 102.07 | 51.03 | 45.51 | $2.20 \times 10^{-14}$ | |
| Cultivar (C) | 1 | 5.75 | 5.75 | 5.13 | 0.0259 | $(\sigma^2 + 8\sigma^2_{TC}) + \kappa^2_C$ |
| Inoculum (I) | 2 | 45.17 | 22.59 | 20.141 | $5.90 \times 10^{-8}$ | |
| T:C | 2 | 0.76 | 0.38 | 0.341 | 0.7121 | $\sigma^2 + 8\sigma^2 TC$ |
| T: | 4 | 3.95 | 0.99 | 0.881 | 0.4785 | |
| C:I | 2 | 0.48 | 0.24 | 0.212 | 0.8092 | |
| T:I:C | 4 | 1.36 | 0.34 | 0.303 | 0.8751 | |
| Residuals | 90 | 100.92 | 1.12 | | | $\sigma^2$ |

* Note: Table is for all fixed effects, but really there are random blocks (=Time) and interactions. In addition, it uses an incorrect error term. We should test Cultivar Mean Sq with Cultivar-Time interaction Mean Sq, but that Mean Sq is less than Residual Mean Sq and only has 2 df.

```
> t.test(Cultivar_3233$Nodules,Cultivar_3234$Nodules,       > anova(lm(Nodules ~ Cultivar, data=noduleSD ))
 alternative = "two.sided", paired = F, var.equal = F)       Analysis of Variance Table

         Welch Two Sample t-test                             Response: Nodules
                                                                        Df  Sum Sq Mean Sq F value Pr(>F)
data:  Cultivar_3233$Nodules and Cultivar_3234$Nodules       Cultivar   1   356.8  356.82    1.74 0.1894
t = 1.3263, df = 133.72, p-value = 0.187                     Residuals 136 27890.0  205.07
alternative hypothesis: true difference in means is not
 equal to 0
95 percent confidence interval:
 -1.580409  8.015137
sample estimates:
mean of x mean of y
 20.85915  17.64179
```

**Figure 1.** Under-fitted model of the data. One-way ANOVA with cultivar (two levels) gives same probability of test of equal means as independent-sample t-test. Cultivar_3233 and Cultivar_3234 stand for the nodule data from cultivar # 3233 and 3234; noduleSD is for the data from both cultivars.

Figure 1 below contains two R programs, one being the anova with a single two-level factor, cultivar, and the other a simple independent-sample t-test for differences of the two cultivars. Both programs output the same results. However, as we mentioned earlier, this analysis is faulty because it does not take into account important contributions to the variance from blocks (Times of greenhouse runs) and inoculum treatment differences. We present these incorrect analyses to illustrate a pitfall and to show equality of the results of the two methods.

**The desired inference requires a proper error term**. The analysis of variance with fixed effects in Table 1 also has the problem that the test of cultivars uses a residual mean square based on 90 degrees of freedom (df), which is not the proper error term for inferring results to new greenhouse environments. Crawley ([11], pp. 173–182) discusses considerations in random vs. fixed effects, and Stroup et al. ([2], p. 10) discuss the scope

of inference when making this choice in models. Plants are nested within each replication or greenhouse run (Time), and to infer results to future greenhouse conditions, we need to assume the greenhouse runs are a random sample from a population of such runs. The error needed for this broader inference should include the Time interaction with treatment combinations. This can be seen in the expected values of mean squares in the analysis of variance table, assuming a random Time-cultivar interaction. (Our example has the anomaly that the cultivar-block mean square is smaller than that of residual even though it is expected to be larger). The main issue is that the residual error term in Table 1 measures plant-to-plant variation, which has inflated degrees of freedom (df). Many researchers mistakenly use such error terms with degrees of freedom (df) higher than warranted by the experiment design, called pseudoreplication by Crawley ([11], pp. 176–182). With mixed model analysis designating the block-treatment interactions as random, some software packages such as JMP automatically perform correct tests of treatment differences for broader inference.

Another learning point about the error term in the analysis of this experiment is the difference in testing with type I and type III sums of squares for cultivars. Type I sum of squares refers to sequential testing of each successive factor in the model and is the default method for R anova. On the other hand, JMP and SAS use Type III sums of squares, which account for all other factors in the model, including interactions. Consequently, it is necessary to consider the default F-tests for cultivar differences when using different software for analyses.

**The data do not fit the usual anova assumptions.** A third issue is that data are counts of nodules on the alfalfa roots, and count data do not usually follow a normal distribution. This leads first to the question of whether to transform the data to solve this issue. Later we also explore using generalized linear models (glm) for analysis. The square root (sqrt) transformation is recommended by Snedecor and Cochran to better approximate the NID anova assumptions ([3], pp. 287–289). Data distributions for original data y and recommended transformation sqrt(y) are given in Figure 2 below. JMP graphically depicts the data distributions, including the original nodule count data and the square root of nodule count. Tests of goodness of fit for normal distributions are below the histograms and normal quantile plots, and the square-root transformation distribution does not significantly differ from the normal distribution.

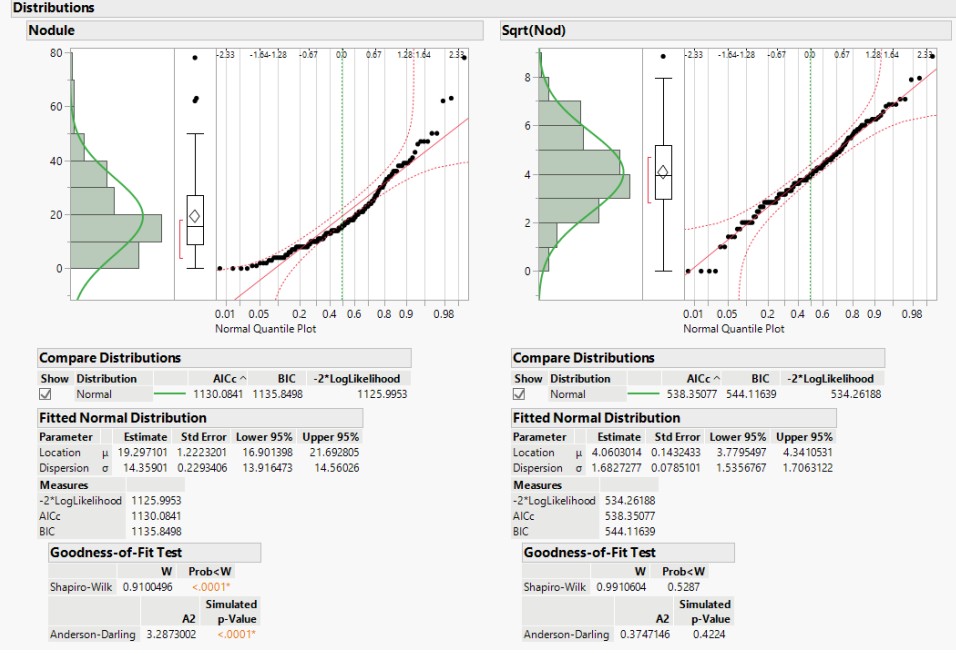

**Figure 2.** JMP distribution (normal quantile plots) and normality tests.

We know that taking averages of the eight plants per treatment combination results in means that are more normally distributed than the original count data. Analysis using averages of the original data is performed with Time (=block factor), cultivar, inoculum, and cultivar-inoculum interaction, given in Figure 3. The same analysis with the averages of square root-transformed nodule counts is also shown. These type analyses are called two-stage analyses [10]. From the output, we see two different estimates of probability of the null hypothesis that cultivars are equal, $p = 0.029$ for the averages of sqrt transformed and $p = 0.134$ for the averages of original data.

**Least Squares Fit**

**Response Mean(Nodule)**

**Whole Model**

**Effect Tests**

| Source | Nparm | DF | Sum of Squares | F Ratio | Prob > F |
|---|---|---|---|---|---|
| Time | 2 | 2 | 1389.5826 | 31.0370 | <.0001* |
| Cultivar | 1 | 1 | 59.5868 | 2.6618 | 0.1338 |
| Rhizobium | 2 | 2 | 438.8470 | 9.8018 | 0.0044* |
| Cultivar*Rhizobium | 2 | 2 | 22.0317 | 0.4921 | 0.6254 |

**Response Mean(Sqrt(Nod))**

**Whole Model**

**Effect Tests**

| Source | Nparm | DF | Sum of Squares | F Ratio | Prob > F |
|---|---|---|---|---|---|
| Time | 2 | 2 | 18.987917 | 63.7348 | <.0001* |
| Cultivar | 1 | 1 | 0.965876 | 6.4841 | 0.0290* |
| Rhizobium | 2 | 2 | 7.008858 | 23.5259 | 0.0002* |
| Cultivar*Rhizobium | 2 | 2 | 0.322661 | 1.0830 | 0.3752 |

**Figure 3.** JMP output for anova using averages of original nodule counts y (**top**) and sqrt(y) (**bottom**).

Using Equation (3) with the sqrt transformation and averages of the eight plants, the cultivars are statistically significantly different (*p*-value = 0.029). More important to report is the 95% confidence interval for the difference between cultivars, for which cultivar 3233 has an average of 3.2 more nodules per root than cultivar 3234, and 95% CI (3.31 to 4.13). This difference is computed by squaring the square-root-transformed cultivar means and subtracting. The standard error of the difference (SED) of means is found by squaring and adding the standard errors of means to obtain the variance of a difference, and taking the square root. However, this is the SED of transformed values, and so must be squared to obtain the SED in the original scale.

**The best methods for analyzing non-normal data continue to evolve.** The generalized linear model in JMP for nodule counts was modeled on fixed Time (greenhouse runs), Cultivar, Rhizobium inoculants, and Cultivar-Rhizobium interaction. This generalized linear model option is found in the upper right corner of the Fit Model window labeled "personality". We use the Poisson distribution with log link and the output is presented in Figure 4. The same analysis in R using the function glm is shown in Figure 5, with the same probability for a test of cultivar differences, 0.000004. These figures indicate very high statistical significance, i.e., a low probability of cultivars having the same average counts of nodules, but this uses an error term with inflated degrees of freedom with the narrow inference restricted to the three fixed greenhouse runs (Times) of this experiment instead of the broader inference across future greenhouse environments. A correct error term for this broader inference uses the random time-cultivar interaction term as specified in expected values of mean squares in Table 1. With this term in the linear R model, we obtain a probability of 0.0259 for the test of cultivar differences, and in the glmm model in Equation (4) this probability is 0.0234 (see the R markdown file).

**Generalized Linear Model Fit**

Response: Nodule
Distribution: Poisson
Link: Log
Estimation Method: Maximum Likelihood
Observations (or Sum Wgts) = 138

**Whole Model Test**

| Model | -LogLikelihood | L-R ChiSquare | DF | Prob>ChiSq |
|---|---|---|---|---|
| Difference | 387.597425 | 775.1949 | 7 | <.0001* |
| Full | 623.526795 | | | |
| Reduced | 1011.12422 | | | |

| Goodness Of Fit Statistic | ChiSquare | DF | Prob>ChiSq |
|---|---|---|---|
| Pearson | 655.6509 | 130 | <.0001* |
| Deviance | 635.7265 | 130 | <.0001* |

**AICc**

1264.1699

**Effect Tests**

| Source | DF | L-R ChiSquare | Prob>ChiSq |
|---|---|---|---|
| Time | 2 | 542.82379 | <.0001* |
| Cultivar | 1 | 21.170995 | <.0001* |
| Rhizobium | 2 | 196.05618 | <.0001* |
| Cultivar*Rhizobium | 2 | 4.9658102 | 0.0835 |

**Figure 4.** JMP output for glm using original nodule counts.

```
> glm_model_number_nodule= glm(Nodules~Cultivar + Inoculum + Time + Cultivar
Inoculum , data = noduelsD, family =poisson(link = "log"))
> summary(glm_model_number_nodule)

Call:
glm(formula = Nodules ~ Cultivar + Inoculum + Time + Cultivar:Inoculum,
    family = poisson(link = "log"), data = noduelsD)

Deviance Residuals:
    Min      1Q   Median      3Q      Max
-6.2078  -1.5117  -0.2341   0.9755   7.7343

Coefficients:
                     Estimate Std. Error z value Pr(>|z|)
(Intercept)          2.786947   0.022891 121.746  < 2e-16 ***
Cultivar1            0.093278   0.020326   4.589 4.45e-06 ***
Inoculum1           -0.419770   0.031712 -13.237  < 2e-16 ***
Inoculum2            0.220510   0.027111   8.134 4.16e-16 ***
Time1               -0.565466   0.035964 -15.723  < 2e-16 ***
Time2                0.589497   0.026718  22.063  < 2e-16 ***
Cultivar1:Inoculum1 -0.005086   0.031711  -0.160   0.8726
Cultivar1:Inoculum2  0.051437   0.027102   1.898   0.0577 .
```

**Figure 5.** R output of GLM with Poisson distribution for nodule counts.

**Summary of the results from different N/R example analyses.** Results of the analyses performed for our example are consolidated in Table 2, which shows the model type, number of data points (either 138 individual or 18 averages for the nodulation data, advantages and disadvantages of each model method, and cultivar *p*-value to illustrate similarities or differences of the software and methods.

**Table 2.** Consolidation of results for models from the nodulation data, including cultivar *p*-values, to show difference in models.

| Model Name | No. DP | Pro | Cons | Cultivar *p*-Value | R/JMP |
|---|---|---|---|---|---|
| Fixed linear model | 138 | Easy; all means and CI available | Not correct error if infer to pop. of runs | 0.0237 | JMP |
| Linear Mixed model | 138 | Random effects | Not best specific distribution (Poisson) | 0.0356 | JMP |
| Avg(Y) linear model | 18 | Easy; all means and CI available | Not best approx. to normal | 0.1338 | JMP |
| Avg(Sqrt(Y)) linear model | 18 | Best normal approximation | Need to back transform to obtain means | 0.0290 | JMP |
| GLM with Poisson | 138 | Best Specific distribution | No random term for broad inference | 0.000004 | JMP |
| Group T test | 138 | For 1 factor | Not for >1 factor | 0.1447 | R |
| One-factor ANOVA | 138 | 1 factor | Normal distribution | 0.1438 | R |
| 3-factor ANOVA | 138 | Can be >1 factors | Normal distribution | 0.0465 | R |
| Linear Mixed Model (lme4) | 138 | Random effects | Not best specific distribution (Poisson) | 0.0287 | R |
| GLM w/Poisson | 138 | Specific distribution | No random term for broad inference | 0.000004 | R |
| GLMM w/Poisson | 138 | Specific distribution | With random | 0.0234 | R |

We note in the green highlighted rows that the glm models for JMP and R have the same *p*-values of 0.000004. The first three R programs in Table 2 illustrate the equivalence of the independent-sample t-test with one-factor anova (*p* = 0.144) and difference with 3-factor anova (*p* = 0.0465). Further, the two JMP models for averages (18 observations) show a difference in *p*-values, 0.1338 and 0.0290 for averages of original data y and sqrt(y), respectively.

**Discussion for Example 1, the Nodules per Root (N/R) Experiment**

One point of emphasis statisticians convey to researchers is to not strictly adhere to a red line *p* = 0.05 for making decisions for testing hypotheses. A more meaningful method is to use confidence intervals for the difference in means of counts for the two cultivars. This difference is an average of 3.7 nodules per root, with 95% CI (3.3, 4.1). We simply use *p*-values here as statistics for comparisons of models.

Several lessons were learned in testing the null hypothesis of equal true average nodule counts for the two cultivars. One is that SAS/JMP generally gives the same results for the same model and assumptions, as does R. This is illustrated in Table 2 with generalized linear model fits of the two software programs. Of course, there may be differences due to different floating-point accuracies of programs, even if the algorithms are the same. Another lesson learned is that the correct error term for testing treatment differences is dictated by randomization in the experiment design, the scope of inference desired (e.g., wishing to infer cultivar differences for a population of potential greenhouse runs), and inherent testing protocols for the particular software used. A third lesson is that assumptions for the analysis can affect *p*-values. For example, averaging nodule counts results in distributions closer to normal compared with original data. The square root transformation recommended by Snedecor and Cochran, in our example, has more nearly normal data than the original observations. Stroup has recommended to use generalized linear models rather than transformations such as the square root [6]. Finally, we use the generalized linear model in JMP and glm in R for nodule counts for the 138 plants. For this paper, these models have fixed effects for Cultivar, Inoculum, and their interaction, but additional analyses consider Time (of greenhouse run) and its interactions as random, leading to mixed models (using GLIMMIX in SAS or glmer in R). Generalized linear mixed models (GLIMMIX in SAS or glmer in R) are helpful for modeling to account for random factors, and the recommended glmm in R in Equation (4) is in the R markdown supplemental file.

Using other modeling distributions, such as negative binomial, can handle over-dispersion relative to the Poisson, whose mean is the same as variance [4,6].

**Example 2: Phosphorous fertilizer application and winter-survival (P_WS).**

**Ex2.1. Objective**. The objective of this experiment is to test whether different phosphorous treatments will increase winter survival of alfalfa plants. The response variable is stand count or survival rate. Response of survival to quantitative phosphorous applied (P), i.e., P-application, for each of two cultivars is to be estimated. Tests of cultivar differences are not a major objective because we already know they differ.

**Ex2.2. Experiment design and conduct.** This is a field plot experiment with three reps whose design may be found in Wang, et al. (2022) [16]. Each plot was thinned to 1000 plants/plot in the fall. Stand count was determined following the winter after phosphorous treatments had been applied. Phosphorous treatments applied the previous year were whole-plot treatments in a split-plot in randomized complete block design with subplots two alfalfa cultivars, one dormant and one semi-dormant. In September, 0, 50, 100, and 150 kg ha$^{-1}$P$_2$O$_5$ (P0, P1, P2, and P3) as calcium phosphate was applied and incorporated into the soil at 5 to 8 cm depth, with irrigation after soil mulching. No harvest was undertaken in the establishment year. Plants were kept well-watered and weed and insect control was conducted as necessary.

**Ex2.3. Data.** These data are an example of a binomially distributed response variable (survival count of $n$ = 1000 plants, see supplemental data "winterSurvival.csv" for detail). The binomial data are a categorical response data example. The survival count for 1000 plants ranges from low of 682 up to 973. Survival rates are correspondingly 0.682 to 0.973. Categorical explanatory factors are "Reps" (blocks of the Randomized Block Design), "Treatments" (the four phosphorous rates randomized on the four whole-plots), and "Cultivars" (two subplot cultivars). The four nominal phosphorous treatments have quantitative phosphorous application rates (P = 0, 50, 100, 150 kg/ha) for estimating survival response to applied phosphorous. Alfalfa yield from the year before the winter survival and its three component cuttings are included in the data set.

The binomial distribution is appropriate for describing the number of successes of a fixed number n of Bernoulli independent trials, such as coin flips with heads or tails. Here, the binary data are success (plant survives) or failure (it dies) of the $n$ = 1000 total plants. The mean, or expected value, of number of plants surviving for the binomial distribution is np and the variance is npq, where p is probability of success and q = (1 − p). For example, if the average proportion surviving is 0.852, the mean number of plants surviving per plot is 852 and the variance is 0.852 × 0.148 × 1000 = 126.096. As you can see, the variance is not independent of the mean, violating the NID assumption. However, for large n values, we can often use the normal approximation to the binomial ([3], p. 130). We have a large enough n (=1000) to meet Snedecor and Cochran's "rule" of np and nq larger than 15 (discussed on page 130 of their book). For the smallest q, we have nq = 1000 × [1 − (973/1000)] = 27, which is larger than 15.

The binomial and Poisson distributions are related. S. D. Poisson derived this distribution ([3], p. 130) for rare events of the binomial distribution by letting n approach infinity as p approaches zero for np, the mean, being constant [3]. For the Poisson, the mean equals the variance, both approaching np compared with the binomial distribution where mean = np and variance = npq. (As n-> infinity and p -> 0, q -> 1, and Poisson variance -> np = mean). Both distributions approach the normal distribution for n being sufficiently large.

A more practical matter for breeders and scientists is to see what aspects of the distributions apply to modelling experiment data. Both distributions, binomial and Poisson, have a relationship of mean to their variance. In the binomial, we often model proportion of successes, p. Binomial has least variance when p is close to 0 or 1 and highest variance when p = 0.5. Poisson is useful for modeling counts and has larger variance for larger counts [4]. A potential problem if we try to model survival count using Poisson and we actually have a fixed number of plants per plot (here, 1000) is that the Poisson distribution has variance continually increasing as survival increases, but binomial survival count variances decrease

as we move closer to 1000. Both distributions have variance determined when we estimate the mean.

In contrast, the normal distribution with variance independent of mean allows estimation of mean and variance separately. This is an advantage because we expect soil and weather error variation in addition to the variance from the relationship with mean of binomial or Poisson random variables. Therefore, it seems reasonable to use a linear model with a separate variance parameter when the normal approximation is justified. Agresti ([4], pp. 75–76) gives examples of the Poisson distribution with overdispersion when any variables affecting the data are missed, and he proposes the negative binomial distribution (with mean $\mu$ and variance $\mu + D\mu^2$ for dispersion parameter D) to model the extra variance (p. 220). He also illustrates generalized linear mixed models with random error structures to better capture this variance in Chapter 10 of the book [4].

Other variables measured in this experiment are total alfalfa yield the season prior to P-treatment and overwintering. Total yield Y = Y1 + Y2 + Y3, the sum of alfalfa yield of the three cuttings.

### Ex2.4. Statistical Analysis

These data analyses are also conducted using R and JMP for comparison, the R codes can be found in the R markdown file "winterSurvival.rmd", and the line by line results from the R codes can be found in the supplemental file "winterSurvival.docx" for details.

We potentially could use total yield or yield for third cutting as a covariate in the modeling of overwinter survival. This variable might be positively associated with survival, indicating healthier plants with higher yield which store more photosynthate in the roots for the winter, or might be negatively related if removal of nutrients from high yield the previous season weakens plants for winter survival. These models are run, but the yield covariate is not statistically significant.

Because of the large sample size (1000), we can analyze assuming the normal distribution (see the criteria quoted above). First, we fit the linear mixed model with JMP as:

$$\text{Survival\_count} = \text{Rep} + \text{Trt} + \text{Rep} * \text{Trt} + \text{Cultivar} + \text{Cultivar} * \text{Trt} \qquad (5)$$

using Treatment (Trt) and Cultivar as fixed nominal factors, and both Rep and Rep-Treatment interaction as random. This way of writing a linear model corresponds to the sequential fitting of factors in a split-plot analysis with whole-plot factor phosphorous treatment (Trt) having whole-plot error estimated as the Rep-Trt interaction.

Regression of survival on quantitative phosphorous applied (P) provides a response curve which can be used to estimate ideal amount of applied P for best winter survival. The 95% CI for optimum rate of applied P is computed from this equation.

The generalized linear mixed model for the binomial variable p = survivalcount/1000 can be fit in R with glmer and model:

$$p = \text{Rep} + \text{Trt} + \text{Rep:Trt} + \text{Cultivar} + \text{Trt:Cultivar} + \text{Plot}, \qquad (6)$$

where Rep, Rep:Trt, and Plot are considered random and the other factors are fixed. The equation refers to modeling the proportion on explanatory factors using the binomial canonical logit link function ([4], p. 67).

**Ex2.5. Interpretation.** The response of winter survival to applied P gives farmers a tool for achieving better alfalfa stands and, potentially, a longer-lasting alfalfa stand. With two cultivars that differ in their inherent survivability, we can see if there is cultivar-P rate interaction, and if not, may give a more general recommendation. If the optimum P-rate for survival differs for the cultivars, different optimum rates could be recommended.

### Example 2 Results of P Fertilizer Application and Winter-Survival (P_WS) Experiment

By using the model in Equation (5) above, the random-effect variance component estimate (from JMP) for Treatment*Rep is estimated as 0, as shown (Figure 6) below.

**REML Variance Component Estimates**

| Random Effect | Var Ratio | Var Component | Std Error | 95% Lower | 95% Upper | Wald p-Value | Pct of Total |
|---|---|---|---|---|---|---|---|
| reps | 0.3566977 | 1061.1458 | 1294.2307 | -1475.5 | 3597.7914 | 0.4123 | 26.292 |
| treatment*reps | -0.195503 | -581.6042 | 909.2068 | -2363.617 | 1200.4084 | 0.5224 | 0.000 |
| Residual | | 2974.9167 | 1487.4583 | 1357.2825 | 10918.474 | | 73.708 |
| Total | | 4036.0625 | 1971.691 | 1868.283 | 14244.792 | | 100.000 |

-2 LogLikelihood = 192.76117801
Note: Total is the sum of the positive variance components.
Total including negative estimates = 3454.4583

**Figure 6.** Summary of the linear mixed model from JMP.

The treatment-rep interaction component is negligible and the residual error term for testing Cultivar and the Treatment-Cultivar interaction, Error (b) based on eight error degrees of freedom (df), is 2974.9. In the split-plot experiment, the Treatment effects are tested with the error (a) term based on six df, the random Rep-Treatment interaction mean square. However, with this component negative, we pool the error (a) and Error (b) terms to obtain a single residual error based on 14 df, which includes all the Rep interactions with other factors. Its estimate is 2476.4. The analysis-of-variance linear model in JMP is Survival_count = Rep + Treatment + Cultivar*Treatment, with results in Figure 7.

**Analysis of Variance**

| Source | DF | Sum of Squares | Mean Square | F Ratio |
|---|---|---|---|---|
| Model | 9 | 132184.92 | 14687.2 | 5.9309 |
| Error | 14 | 34669.58 | 2476.4 | Prob > F |
| C. Total | 23 | 166854.50 | | 0.0017* |

**Effect Tests**

| Source | Nparm | DF | Sum of Squares | F Ratio | Prob > F |
|---|---|---|---|---|---|
| reps | 2 | 2 | 20601.750 | 4.1596 | 0.0382* |
| treatment | 3 | 3 | 19261.500 | 2.5927 | 0.0940 |
| cultivars | 1 | 1 | 91760.667 | 37.0541 | <.0001* |
| cultivars*treatment | 3 | 3 | 561.000 | 0.0755 | 0.9722 |

**Figure 7.** Summary of the ANOVA from JMP.

The anova has statistically significant rep and cultivar differences, and the test of phosphorous treatment differences has a probability of 0.09. Because our objective is to find the survival relationship with applied P, it is better to model the response to the quantitative Phosphorous applied than to test statistical significance. The treatment-cultivar interaction is practically nonexistent, and that term is omitted from the model.

The three degrees of freedom for treatments can be examined using three orthogonal contrasts whose sums of squares (SS) will add to the treatment SS in Figure 7 above. We fit the linear, quadratic, and cubic orthogonal contrasts using JMP, and results are shown in Table 3. These use the same error term as in the anova Figure 7, and their SS add to 19,262, the same as the three-df treatment SS in Figure 7. The tests of significance for linear, quadratic, and cubic are given in Table 3 and use our best estimate of error. As stated previously, do not put too much emphasis on significance tests and a "red line" at a probability of 0.05. Here, our principal objective is to fit survival response to P-application. The cubic term is not necessary, so we fit a quadratic response curve.

**Table 3.** Summary results of the three contrasts.

| Contrast Type | Sum Square (SS) | Num DF | F Ratio | Prob > F |
|---|---|---|---|---|
| Linear Contrast | 8267 | 1 | 3.3382 | 0.0891 |
| Quadratic Contrast | 10,584 | 1 | 4.2739 | 0.0577 |
| Cubic Contrast | 411 | 1 | 0.1658 | 0.6900 |
| Total Treatment | 19,262 | 3 | | |

Without treatment-cultivar interaction, we can model survival on quantitative P for both cultivars. As seen in Figure 8 below, the quadratic fit to the data for cultivars combined is:

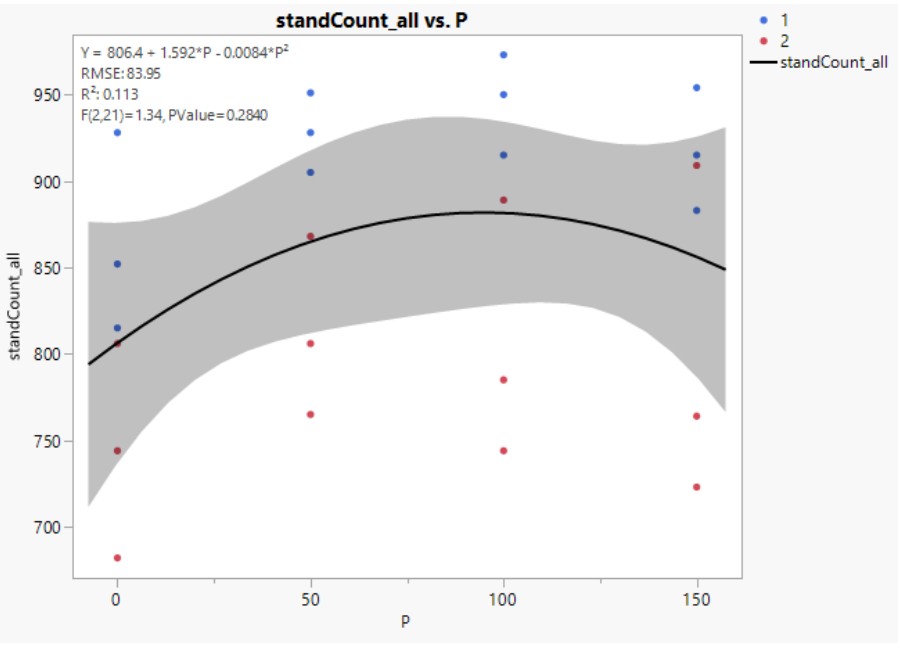

**Figure 8.** Quadratic regression line between the fertilizer levels and stand counts. The blue dots are from cultivar number 1, and the red ones are for cultivar number 2. The letter P in the equation is for the phosphorous rate in kg/ha.

Y (Survival) = $806.4 + 1.592P - 0.0084\ P^2$, where P is kg/ha applied $P_2O_5$. Standard errors for the coefficients of P and $P^2$ are 0.569 and 0.0036, respectively.

To find the optimal rate of P to apply, we set the first derivative of Y with respect to P to zero, $1.592 - 0.0168\ P = 0$, and solve for the optimum, which is P = 94.8, or 95 kg/ha applied P as $P_2O_5$. The estimated standard error of the optimum P rate is 1.325, and a 95% confidence interval for the optimum is (92.0, 97.6).

The figure from JMP shown below has a correct equation in the key in the upper left, but does not have a proper root mean squared error (RMSE) or significance test. The model fits the equation correctly but does not remove recognized variance sources reps and cultivars, inflating the error and rendering *p*-values too high. The correct anova error variance is 2476.4, with RMSE = 49.8. The linear and quadratic contrast probabilities given in Table 3 above are correctly computed, with 0.09 and 0.06 as *p*-values for linear and quadratic, respectively. The cubic term is lack of fit for the quadratic equation and is not significant, with P = 0.69.

The model to fit response to applied P uses the quantitative variable P and quadratic term $P^2$. This is undertaken for cultivars together (the cultivars had similar response curves). The result is graphed in Figure 8 below.

**Summary of the results from different P_WS analyses.** Results of the analyses performed for our example are consolidated in Table 4, which shows the model type, number

of data points (24 for the winter survival data), advantages and disadvantages of each model method, and cultivar P values to illustrate similarities or differences of the methods.

**Table 4.** Consolidation of results for models from the winter survival data, including cultivar *p*-values, to show the difference in models.

| Model Name | No. DP | Pros | Cons | Cultivar *p*-Value |
|---|---|---|---|---|
| LM w/split plot | 24 | Familiar LM, good approximation, accounts for error structure | Normal distribution assumption needed | 0.0875 |
| GLM w/binomial | 24 | Specific family distribution | No random error; Using inflated df | $5.90 \times 10^{-15}$ |
| GLMM w/binomial | 24 | Specific family distribution accounts for error structure | Hard to explain | 0.0311 |

**Discussion for Example 2, the P Fertilizer Application and Winter Survival (P_WS) Experiment**

The R code for analysis and updates to this analysis is contained in the R markdown file "winterSurvival.rmd". We compare models with R analyses and results for the Gaussian linear model, for a generalized linear model (glm) using a binomial distribution with logit link, and finally for the generalized linear mixed model glmm with random field variance components for Reps and the Rep-Treatment interaction using binomial distribution modeling. There are differences in *p*-values for the test of Treatment mean differences for these methods. The standard tests for different scopes of inference are dictated by the model fitting of, for example, the random Rep-Treatment component which has correct error df for the broader inference (over future reps for which this set is a random sample). Fang and Loughin used a simulation for a split-plot design and found methods based on linear mixed models and generalized linear mixed models held Type I error rates better than generalized linear models [17].

We see many of the same principles illustrated in this binomial data example as were shown in the Poisson nodule count example, such as the need to pay attention to the scope of inference and to the error terms for correctly testing hypotheses in deciding fixed versus random effects ([2], p. 12). Such principles are a main point of emphasis for these examples.

We have seen that there are several ways that statistical models can be specified, some more true to the underlying distributions, and others which are approximations but may have more familiarity to the data analyst. For this winter survival example, the normal approximation turns out to be very good, and is straightforward to use for those trained in the use of linear model methods, such as regression and analysis of variance. The survival count data may at first seem to be an example in which the Poisson modeling for count data seems appropriate, but the experiment is to count surviving plants from an initial stand of 1000. This makes the underlying distribution binomial, which differs from the unbounded count data which have a mean equal to the variance. In fact, as we discussed earlier, the closer the counts are to 1000 the less variance they have. The binomial distribution seems a much better option to match the actual conditions of this experiment.

A second consideration in modeling these data is the actual experiment design and spatial variation generating what we assume are random error terms. The (P_WS) experiment was arranged as a split plot with phosphorous treatments as whole plots and cultivars as subplots. This subsumes an error structure with one error term for whole plots and another for the subplots. This leads to random error designations as discussed by Crawley ([11], pp. 173–184). The generalized linear models, even with random terms (glmms), would need the random error terms corresponding to those of the linear models which assume normal distributions. The normal distribution model has a variance parameter as well as

the parameter for mean, and modeling using glms and glmms is often performed with some provision for this over-dispersion that is inherent when we assume variance is only dependent on the mean. Although this extra variance can be modeled with a distribution such as negative binomial having an extra dispersion parameter, our approach is to include a random plot variance to account for the residual field variance and random phosphorous Trt-Rep interaction for the whole-plot error variance. As stated already, the methods for these CDA analyses are continually evolving.

## 4. Summary of Analysis Approaches and Models for Both Examples

The overall analysis approach starts with the experiment objective, which, along with the experiment design, guides the model's development. For the N/R example, the greenhouse runs (Times) are the blocks, and cultivars and inoculum are treatments. Equation (2) is a linear mixed model with the NID assumptions. It is important to check the data for NID assumptions, e.g., checking distributions using histograms and testing goodness of fit to a normal distribution, which can be achieved using either R or the JMP analysis distribution platform. Count data from this experiment are not expected to be from a normal distribution and are generally modeled as Poisson. The square-root transformation allows for more accurate probability statements than just the normal approximation. Averages of individual plants result in data closer to a normal distribution via the central limit theorem, and in our nodules per root experiment, means of the original data are not as close to a normal distribution as means of square root transformed data. The second model, Equation (3), is a better normal approximation than Equation (2). A disadvantage of the sqrt transformation approach in Equation (3) is the need to back-transform to obtain means and 95% confidence interval endpoints in the scale of original counts.

Modeling CDA with a generalized linear model better captures the underlying Poisson distribution, and glm results are in Table 2. However, random error terms not present in the glm are important for proper inferences to achieve the objectives of this experiment. In the N/R example, *p*-values for the test of cultivar mean differences are much lower when fixing the scope of inference to the specific three greenhouse runs (Times) than for a broader inference in which a random Time-cultivar interaction was included. This broader scope of inference is for a population of potential future greenhouse runs, and we determined the 95% CI for cultivar difference for this broad inference. Finally, our glmm model in Equation (4) contains the important treatment terms and models random variation with normal distributions having zero means and variance parameters to be estimated for greenhouse pots and for Time. The results of the R program for Equation (4) are in the R markdown Supplementary Materials, and the *p*-value for testing cultivar differences, 0.0234, is not much different from the cultivar *p*-value of 0.0290 from the JMP model run for Equation (3). In the P_WS example, experiment design factors with fixed effects and field variation with random effects are modeled. The linear mixed model, Equation (5), includes the split plot random field effects of replication and replication-phosphorous treatment interaction, which is the whole-plot error term. For this P_WS example, the normal approximation is very good, and using quantitative applied phosphorus, we could estimate the response as a quadratic equation. This allowed estimation of an optimum phosphorus application rate. Equation (6) is a glmm with the same random field effects as the normal linear model and an additional random normal effect for the field plot because the generalized linear equation for binomial with logit link does not have the random error variance parameter. Some authors, such as Stroup [6], recommend fitting such a glmm with a negative binomial. In this example, we decided to use the well-recognized normal distribution for plot error. A drawback of the glmm model in Equation (6) compared with the approach for Equation (5) is the lack of the assumption of additive treatment effects ([3], pp. 282–286) when means are related through the glmm link function, in this case the logit. Data scientists are much more familiar with the whole-plot and split-plot error terms of a linear model and the ability to decompose phosphorous treatment sums of squares into

orthogonal contrasts and fit a quadratic equation to applied phosphorous to estimate the optimum rate for plant survival.

## 5. Conclusions

Takeaways from the statistical analysis are that SAS or JMP and R give essentially the same results for the same model with the same assumptions and error terms. The simple, independent-sample t-test will not agree with the anova when more than one factor is in the model.

The N/R counts per root is an example of Poisson data. The objective is to determine the cultivar mean difference. A generalized linear model with fixed time-cultivar interaction has high significance with low *p*-value for cultivar difference, but using a random time-cultivar interaction error term and averages of the square-root transformed data results in the broader-inference 95% confidence interval for cultivar difference in potential future greenhouse environments.

In the P_WS binomial example, the objective is to relate phosphorus to winter survival, knowing that the two cultivars have different winter survival. The 1000 plants per plot render the normal distribution approximation good, and a linear model fits the split-plot error structure. A quadratic equation incorporating this structure allows estimation of the optimum rate of phosphorus to apply.

Methods and software for modeling CDA are continually evolving. The generalized linear regression models are currently preferred to former methods involving testing with normal distribution assumptions. Proportion data are almost always modeled as binomial and count data are generally modeled as Poisson, but modeling may also be performed as negative binomial because of the over-dispersion problem. Generalized linear models can account for additional variation from greenhouse pots or field plots.

Our purpose in this article has been to raise the awareness of the reader of the existence of categorical data, i.e., recognition of binomial and Poisson distributions in their measured data, and to illustrate newer methods for analysis available in R and JMP. Potential pitfalls in analyses are highlighted to emphasize points that have long been recognized but need to be continually reviewed. The two example data sets using alfalfa are not atypical of what crop breeders and agronomists would see in their experiments and help elucidate categorical data analysis methods as well as stress good statistical practices in the analysis of agronomic data.

Finally, it is important to consult with experts for help in statistical analyses. For example, utilize the expertise of a statistician at your university or place of work to ensure you are using the best statistical methods.

**Supplementary Materials:** The following supporting information can be downloaded at: https://www.mdpi.com/article/10.3390/crops2020012/s1. Supplementary Material S1: two data sets (nodules.csv and winterSurvival.csv), two R markdown files (nodules and winterSurvival.rmd), and two word documents (nodules.docs and winterSurvival.docx) containing the line by line R code and their outputs form the R scripts for repeatability.

**Author Contributions:** Conceptualization, R.P.M. and Z.X.; methodology, R.P.M.; software, R.P.M. and Z.X.; validation, R.P.M. and Z.X.; formal analysis, R.P.M. and Z.X., investigation, B.B., Y.C., D.A.S. and Z.X.; resources, D.A.S.and Z.X.; data curation, Z.X.; writing—original draft preparation, R.P.M.; writing—review and editing, R.P.M. and Z.X.; visualization, R.P.M. and Z.X.; supervision, D.A.S.; project administration, D.A.S.; funding acquisition, Z.X. and D.A.S. All authors have read and agreed to the published version of the manuscript.

**Funding:** This work was supported by the USDA-NIFA-AFRP (2014-70005-22543) and USDA ARS in-house project 5062-12210-004-000D.

**Data Availability Statement:** The original two data sets, and the corresponding two R markdown files are available for download as supplemental materials in the journal website. They are also available by request from the corresponding authors.

**Acknowledgments:** This paper is a joint contribution from the Plant Science Research Unit, USDA-ARS, and the Minnesota Agricultural Experiment Station. Mention of any trade names or commercial products in this article is solely for the purpose of providing specific information and does not imply recommendation or endorsement by the US Department of Agriculture. USDA is an equal opportunity provider and employer.

**Conflicts of Interest:** The authors declare that there is no conflict of interest regarding the publication of this article.

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
