# Peer review of "Good Statistical Practices in Agronomy Using Categorical Data Analysis, with Alfalfa Examples Having Poisson and Binomial Underlying Distributions"

_2673-7655, doi:10.3390/crops2020012_

Round 1

Reviewer 1 Report

While I appreciate the general idea and objective behind this work, the manuscript requires a lot of work. The manuscript lacks structure and the language would in general benefit from being “tightened up”. To my taste, at least, it is too informal and resembles spoken language. I mention some specific suggestions below, but it is a general issue throughout the manuscript. The line of thinking becomes really hard to follow when details are left out and methods are not properly introduced before results are compared.

General comments

The introduction is missing some background. As it is written now, it is mainly an abstract.

A large part of section 2, the literature review, is actually background and would fit better as such being part of a proper introduction.

The fourth objective does not really match with the rest of the story, the result section, the discussion, or the conclusions.

In-text citations are a mix of numerical references and author year references.

The reference list needs an update to include all relevant information for all references, i.e., volume and page numbers.

In general, there is a lack of putting things into context. Several times the authors assume the reader has pre-knowledge on the topics/examples mentioned, e.g. discussing experiments never introduced or findings from papers where the content is not known. 

Most of the numbers points in the literature review do not fit into the story/objectives of the manuscript.

Please add a thorough description of each of the statistical models/ analysis strategies compared before giving the results.

Why give examples of code for analyses that are not correct to use? (e.g. Figure 1) And if showing code, please show enough code for it to make sense. In figure 1, the model “model_nodule” is not shown why the anova(model_nodule) command basically makes no sense.

There is a problem with the use of probability vs p-value which seems to refer to the same thing in the manuscript. Please consider revising "probability" to "probability of ..." or simply "p-value for the test of ..."

All graphs and most tables should be modified for better readability. Output from analyses could be presented in a nicer format than a copy of the output from the software.

Typically, two-stage analyses require using the standard errors from the first step in the second step as well.

Why not show a generalized linear mixed model, which to me would be the right choice of analysis in example 1.

There is a problem with the section numbering, please revise.

There is a result section and a discussion after example 1, but either after example 2. I recommend adding a joint discussion after the second example before the conclusion.

A large part of the conclusion does not build on the results from the present work. Actually, I would consider a major part of the conclusion to be a discussion rather than a conclusion.  

Specific comments

  1. 37: Is something missing? The purpose/objective of this article…
  2. 43: Suggest revising to: “… with an outline of the approach…”
  3. 71: Suggest changing “normal” to “normal distribution”. Here and elsewhere
  4. 72: suggest revising to: e.g. “e.g. winter-survival yes/no …”
  5. 74: suggest using another word than “ridiculous”
  6. 75: “Agesti also states…” Agesti was not mentioned before this
  7. 86: This sentence only makes sense if the proportion holds across different treatments.
  8. 87: CDA is already defined
  9. 89: You don’t get a variance independent of the mean simply by transforming your data. You get a different relationship between variance and mean.
  10. 104: binomial data?, binomial distribution?
  11. 117: Here you refer to an N/R experiment that has not been introduced. Either give a proper background of this experiment or refer to the findings in more general terms.
  12. 142-143: I have a hard time getting what is the purpose of this sentence.
  13. 144-145: Suggest revising this sentence. What do you mean by “nested designs for R”?
  14. 151: Why is it only helpful for R?
  15. 191-197: If the objective of this manuscript is to report the findings of the two experiments, more details are needed, e.g. time of year, pot size, cultivars, nodulation strains, etc.
  16. 195: What was the reason for the six missing plants? This may be important for the analysis and interpretation of results.
  17. 211: When first defined the abbreviation was glms. Please be consistent.

Table 1: ANOVA output from which program? And from which model? The output does not resemble what was described in the statistical analysis section. What is NTB_random? “Cultivar p-value differ from t-test value”, please revise.

  1. 249: There is no general agreement on what needs to be added as random effects, so please add a reference for this statement.
  2. 290: personality?
  3. 294: “probability of greater chisq”, please revise.
  4. 350: Is P defined previously? There seems to be an unnecessary shifting between using P and using phosphorous treatment.
  5. 381: “their”? who?
  6. 422: It is a very unusual way of writing up a linear mixed model with fixed and random terms mixed.
  7. 441: What exactly is meant by “we fit the model to data from both cultivars”?
  8. 454: Please also report the standard errors for the estimates.
  9. 456: Please also report SE or confidence interval on the optimal value of P.
  10. 475: “output above”, please be more specific and refer to tables or figures by number.

Author Response

Dear reviewer, 

We appreciate your comments, highly regard their recommendations, and have acted upon your suggestions. We have carefully made major revisions to the manuscript according to the two reviewers' recommendations.

We started with changes and responses for Reviewer 2 (R2) and then responded to Reviewer 1 (R1). Revisions were done first to incorporate the formatting, numbering, and structural changes suggested by R2 to help readers better navigate the paper. The revisions also make the presentation format of the two examples the same.

We have finished the following three requirements and submitted them to the system. Please let us know if you need any additional information for the revised submission.

  1. A cover letter with a list of changes or reasons for keeping original for each point raised in the review
  2. A separate revised manuscript file with highlighted changes
  3. A cleaned revised manuscript without tracked changes

Thank you and best regards,

Reviewer 2 Report

Review of "Good statistical practices in agronomy using categorical data 

analysis, with two alfalfa examples"

  The manuscript is devoted to a very important topic, and one that attracted relatively less attention. As the authors note, the methods are continually evolving and thus it is good to summarize best practices regularly. The actual focus of the paper: is" modifications when ... underlying distributions are Poisson or binomial". Authors highlight the need for these analyses and present various pitfalls. Example 1 presents several approaches and a useful discussion of issues with each approach. In general the paper is well written and enjoyable, however the formatting is lacking.    

Issues:

In Example 1 author's message seems incomplete - what is the best approach for this example? Authors do note that "A correct error term for this inference uses the random time-cultivar interaction term" however the authors did not include mixed model approach into the table ( glmer analysis was only mentioned in Discussion but not shown). If that approach is their final advice, it would be nice to include it into the comparison table explicitly and perhaps discuss potential pitfalls.

Example 2 is done using normal approximation. For complete comparison, similarly to example 1, it would be good to say how results would be different if instead of normal approximation, a logit transformation is used first, and also present a complete analysis via binomial GLM (or GLMM).

Other minor issues:  

Only one type of data is actually considered - count data (second example is a proportion derived from count data). The generality of the title made me expect that examples with nominal or ordinal response variable will be also included. The paper is good already, but perhaps the title could be more specific .

The structure of the paper is non-conventional (each of the two examples is structured as a sub-paper, with its own Results and Discussion). This leads to some difficulty in reading, at least initially. I agree that this division may be indeed better than presenting two examples in parallel through methods, results, and Discussion etc. However, it would be good to format the text in a way that makes it more clear from the beginning.   

The start of Example 2 is not highlighted in any way. It is difficult to navigate the paper.   The format of the presentation of the two examples is not the same . Also, the numbering is confusing  (3 - Results, 4 - DIscussion, then Example 2 with its own numbering)  

The literature review seems a bit small for a topic like this.  References to recent analysis packages can be added.   

Line 148: Link  http://goanna.cs.rmit.edu.au/~fscholer/anova.php is broken - the resource is not available any more.

Author Response

(The authors gave the same response as above.)

Round 2

Reviewer 1 Report

The authors have made revised the manuscript according to the suggestions. However, I still have some suggestions for improvements.

The manuscript would still benefit from proofreading.

The first part of the introduction still resembles an abstract or school assignment or report

Data should be plural

The mix between writing models for R and JMP may be a bit confusing for readers not familiar with any of the two. I would suggest presenting the models in general (not software-specific) terms. Actually, most of the models presented hold for both software programs anyway.

Please, present p-values as decimal numbers, not as scientific numbers.

Most importantly, the manuscript is lacking a more consistent walk-through/ comparison of analysis approaches.

Parts of the conclusion belong to the discussion

Author Response

Response to Reviewer 1, Round 2

Thank you for the very good suggestions and comments. Our answers to your issues are presented in red in the text below. 

--  Ronald P Mowers and Zhanyou Xu

Comments and Suggestions for Authors

The authors have made revised the manuscript according to the suggestions. However, I still have some suggestions for improvements.

The manuscript would still benefit from proofreading.

Reply: The manuscript has been proofed again.

The first part of the introduction still resembles an abstract or school assignment or report

Reply: The introduction has been modified by changing several sentences (lines 42 – 67 in the original) and we have changed from using first person to make it less like an abstract or school report.

Data should be plural 

Reply: This has been changed in lines 18 and 598 of the original.

The mix between writing models for R and JMP may be a bit confusing for readers not familiar with any of the two. I would suggest presenting the models in general (not software-specific) terms. Actually, most of the models presented hold for both software programs anyway. 

Reply: We appreciate reviewer’s suggestion to make general for readers not familiar with any of the two methods (R and JMP).  The model nomenclature is very similar for both R and JMP, with only the interaction differing between the two (JMP uses asterisk “*” and R uses colon “:” for interaction terms). We also want to target analysts for crop research and let them practice the R codes in the supplemental files. The model style is explained in Ex1.4. Although the glm notation is as presented in the computer programs, it is, for glms and glmms, the equation relating the mean via a link function, which is now explained more thoroughly. Our decision not to use all Greek alphabet letters for fixed effects and Latin for random effects, as done by Agresti, is to allow better readability by agronomists and plant scientists and better correspondence to the variable names in the experiments.

Please, present p-values as decimal numbers, not as scientific numbers. 

Reply: This was done completely in the text, in lines 349 and 372 and in Tables 5 and 9. However, values less than 10-7 in tables still are in scientific notation.

Most importantly, the manuscript is lacking a more consistent walk-through/ comparison of analysis approaches. 

Reply: We added a summary of analysis approaches and models for both examples to give a consistent walk-through and comparison of the approaches. This also addresses the suggestion on round 1 of the review for Reviewer 1 that the two examples should have some common discussion. We did use some of what was formerly in the Conclusion section in this Summary.

Parts of the conclusion belong to the discussion  

Reply: The Conclusion section has been rewritten. Some of what was formerly in the Conclusion section is now in the new Summary. Conclusion paragraphs on each example were put into this section.